# 🐟 R²-CoD: Understanding Text-Graph Complementarity in Relational Reasoning via Knowledge Co-Distillation

**Zhen Wu    Ritam Dutt    Luke M. Breitfeller**
**Armineh Nourbakhsh**[*]  **Siddharth Parekh    Carolyn Rosé**
Language Technologies Institute, Carnegie Mellon University
{zhenwu, rdutt, mbreitfe, anourbak, spparekh, cprose}@andrew.cmu.edu

## Abstract

Relational reasoning lies at the core of many NLP tasks, drawing on complementary signals from text and graphs. While prior research has investigated this dual complementarity, a detailed and systematic understanding of text-graph interplay and its effect on hybrid models remains underexplored. We take an analysis-driven approach using a unified architecture with knowledge co-distillation (CoD) across five diverse relational reasoning tasks. By tracking how text and graph representations evolve during training, we uncover interpretable patterns of alignment and divergence, and provide insights into when and why their integration is beneficial.

## 1 Introduction

Incorporating modalities beyond the surface form of the text has shown promise for challenging natural language processing (NLP) tasks like **relational reasoning** based tasks, which require understanding or inferring the semantic relationships within the input [Nastase et al., 2015]. Key examples include relation extraction [Christopoulou et al., 2019, Guo et al., 2020], knowledge base question answering (KBQA) [Tian et al., 2024, Feng and He, 2025, Gao et al., 2025], and structured document interpretation or reasoning [Wang et al., 2023, Chen et al., 2025].

A common and effective way to encode relational structure is through graphs [Yao et al., 2018, Lee et al., 2023, Gururaja et al., 2023, Dutt et al., 2022], where nodes represent textual units and edges encode relationships [Scarselli et al., 2009, Bruna et al., 2014, Veličković et al., 2018]. This explicit structure supplies complementary signals often absent from plain text.

While many tasks utilize this text-graph representation to improve performance, how they complement each other remains underexplored. Prior work notes that models often fail to integrate modalities effectively [Stanton et al., 2021], raising open questions: How do text and graph representations relate to each other during learning? Under what conditions is their integration beneficial?

We address these questions with an analysis-oriented approach and introduce a unified framework for characterizing the alignment and complementarity between text and graph representations under knowledge co-distillation (CoD) [Yao et al., 2024]. Across five diverse relational reasoning tasks, we systematically analyze how text and graph representations complement each other under CoD, identify consistent patterns ranging from complementarity to alignment, and provide practical insights to inform the effective use of CoD.

---

[*] Work done while the author was a student at Carnegie Mellon University.

## 2 Task suite and formulations

We select five relational reasoning tasks to span a spectrum from strong complementarity to near-complete alignment between text and graph representations (**Figure 1**). The tasks vary in whether the graph encodes the prediction target explicitly, whether nodes correspond directly to text spans, and whether reasoning is local or global. The tasks include event temporal relation extraction (ETRE), multilingual relation extraction (MLRE), form understanding (FU), question answering over knowledge bases (KBQA), and relation path prediction (RPP). We outline the details in Table 2.

## 3 Unified framework for analysis

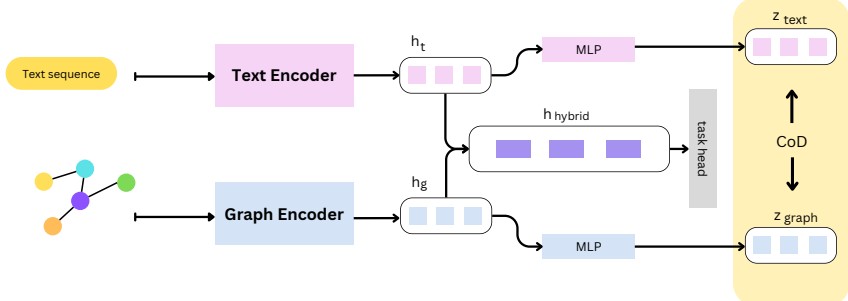

Figure 2: Unified framework for analyzing text-graph complementarity. A text sequence and its graph are encoded separately, then (1) combined for task prediction and (2) projected into a shared space, where a contrastive co-distillation (CoD) objective promotes mutual learning and enables representation-level analysis.

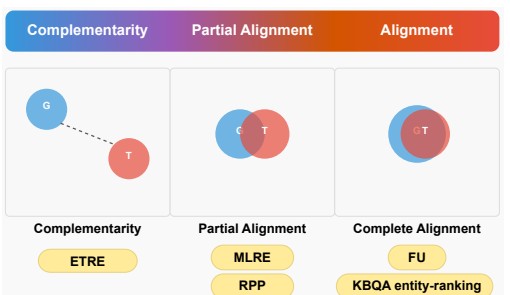

Figure 1: Task spectrum of representation relationships from complementary to alignment.

We propose a unified, task-agnostic framework, $R^2$-CoD (**Figure 2**), to analyze how text and graph representations relate during learning.

Across tasks, each instance corresponds to a text-graph pair. These are encoded using modality-specific encoders: $h_t = f_t(q)$ and $h_g = f_g(G)$. We then create a hybrid representation $h_{\text{hybrid}}$ through concatenation or residual connection to perform task-specific prediction and compute the task loss: $h_{\text{hybrid}} = f_{\text{fuse}}(h_t, h_g)$, $\mathcal{L}_{\text{task}} = \mathcal{L}(h_{\text{hybrid}}, y)$, where $y$ denotes the gold supervision and $\mathcal{L}(\cdot, \cdot)$ is the task-specific loss function. We present model configurations, loss function, and evaluation metrics used for each task in Table 5. To enable direct comparison, we map each representation into a shared latent space via MLP projection heads during training: $z_{\text{text}} = \text{MLP}_t(h_t), z_{\text{graph}} = \text{MLP}_g(h_g)$.

While learning a shared space enables comparison, it cannot solely influence how text and graph will complement one another. We thus apply a contrastive knowledge co-distillation (CoD) objective [Yao et al., 2024] which combines a contrastive loss with a stop-gradient operation [Chen and He, 2021] to explicitly encourage bidirectional knowledge transfer.

Formally, the contrastive loss $l_{cl}$ between the teacher $t$ and the student $s$ representations is:

$$l_{\text{cl}}(t, s) = -\log \frac{e^{sim(t,s)/\tau}}{\sum_u \mathbb{1}_{[u \neq t]} e^{sim(t,u)/\tau}} \quad (1)$$

where $u$ indicates representations from the training data other than $t$ and $s$, $sim(.,.)$ is cosine similarity, $\tau$ is the temperature scaling parameter [Tian et al., 2022]. This bidirectional design ensures that either modality can act as teacher or student at each step, thus mutually distilling knowledge

from each other. Hence, the full CoD loss is computed as

$$\mathcal{L}_{\text{CoD}} = \frac{1}{2} \sum_i [l_{\text{cl}}(z_i^{\text{text}}, \hat{z}_i^{\text{graph}}) + l_{\text{cl}}(z_i^{\text{graph}}, \hat{z}_i^{\text{text}})] \tag{2}$$

where $\hat{\cdot}$ is the stop gradient operator [Chen and He, 2021] that sets the input variable to a constant. Finally, we combine this with the task loss to enable end-to-end model optimization: $\mathcal{L}_{\text{total}} = \mathcal{L}_{\text{task}} + \lambda \mathcal{L}_{\text{CoD}}$, where $\lambda$ controls the weight of the CoD signal. CoD serves as a task-agnostic framework to facilitate learning and analysis over dual modalities.

We assess representational relations using PCA visualizations [Ferrone and Zanzotto, 2020] as well as cosine similarity and within/between-modality distances.

## 4 Analysis and discussions

### 4.1 RQ1: Does combining text and graph representations improve performance?

We compare four model configurations: (1) text-only, (2) graph-only, (3) hybrid with CoD, and (4) hybrid without CoD in Table 1 and Table 8. Across the tasks and model backbones, we observe that hybrid models consistently outperform the text-only and graph-only baselines, with CoD leads to further gains at minimal computational cost (Table 7). The only exception is MLRE where the hybrid approaches achieve performance comparable to the the text-based baseline, possibly because large-scale pretraining enables transformers to

| Task | Dataset | Text | Graph | CoD | T+G |
|------|---------|------|-------|-----|-----|
| ETRE | TDDAuto | 61.6 | 34.6 | **77.1** | 68.9 |
| FU | FUNSD | 33 | 22 | **38** | 35 |
| MLRE | RED$^{\text{fm}}$ | **79.7** | 48.6 | 78.6 | 79.5 |
| RPP | WebQSP | 62.4 | 63.2 | **65.9** | 65.6 |
| KBQA | WebQSP | 80.7 | 52.2 | **83.8** | 83.5 |

Table 1: Task performance. Best performance in **bold**, second-best underlined.[2]

encode syntax internally [Starace et al., 2023, Liu et al., 2024], thus employing off-the-shelf parsers to capture dependency information shows little promise [Sachan et al., 2021]. In KBQA, where text and graph encode the same information in different formats (linearized vs. topological), CoD offers only marginal gains. In contrast, tasks like FU where text and graph encode different information (form content versus layout structure from OCR), CoD shows more improvement.

### 4.2 RQ2: How do text and graph representations relate during learning?

**Complementarity (ETRE):** The text and graph representations remain well-separated throughout training (Figure 3a and 8). In ETRE, the text representation provides local semantic cues around event mentions, while the graph encodes structural information in an attempt to quantify semantic temporal and discourse relations. These structural and semantic divergences could lead text and graph representations to retain independent representation space.

**Partial alignment (MLRE and RPP):** Here, the text and graph representations move closer in the shared space during training, yet do not collapse into a single unified cluster (Figure 3b, 10 and 13). This behavior aligns with the task objective, for example, in RPP, the objective is to classify the reasoning path traversed in the graph, not specific tokens or nodes. Thus, the representations can evolve in parallel without needing to fully align.

**Complete alignment (FU and KBQA):** Here, text and graph representations show strong convergence (Figure 3c and 16). By the final epochs, the paired embeddings often form overlapping clusters. The fine-grained one-to-one correspondence between a graph node and a text token span likely encourages representations to align. In FU, OCR tokens are linked to spatially grounded nodes, while in entity ranking, candidate answer entities are matched between graph nodes and text tokens.

Cosine similarity increases due to CoD, but tasks with complementarity like ETRE show a weaker increase (bounded near 0.4). Complementarity is reflected when between-group distance stays higher

---

[2]We present results for one representative dataset per task due to resource constraints. Similar trends hold for other datasets. For FU, the model was pretrained on a 1,000-example subset of its original pretraining corpus.

than within-group distance, as in ETRE. In partial and complete alignment tasks, their between-group trends diverge: it increases in partial alignment tasks (MLRE and RPP), whereas it decreases steadily and eventually approaches the within-group distances in complete alignment ones (FU and KBQA).

### 4.3 RQ3: How do task characteristics shape the effects of CoD?

**Same input, different task objectives**    Although RPP and KBQA share the identical input, they differ in task objectives: RPP identifies graph-level reasoning patterns, and KBQA scores entities at a node level. Under CoD, RPP shows partial convergence whereas KBQA aligns strongly, which suggests that the level of reasoning (global vs. local) shapes representational behavior.

**Same reasoning scope, different graph construction:**    ETRE (complementarity) and FU (complete alignment) both involve pairwise reasoning, but their graph designs differ in how directly they capture the task. FU graphs encode layout relations that closely match the target key–value associations. In ETRE, the graph encodes linguistic cues that support but do not directly define the target temporal relation. This indicates that how well the graph structure reflects the task objective can influence whether CoD promotes complementarity or alignment.

**With or without token-node correspondence**    In FU and KBQA entity-ranking, there exists a one-to-one correspondence between graph nodes and text token spans, unlike the complementary information encoded in ETRE. This highlights that explicit token–node correspondence could act as a structural prior that facilitates CoD towards alignment.

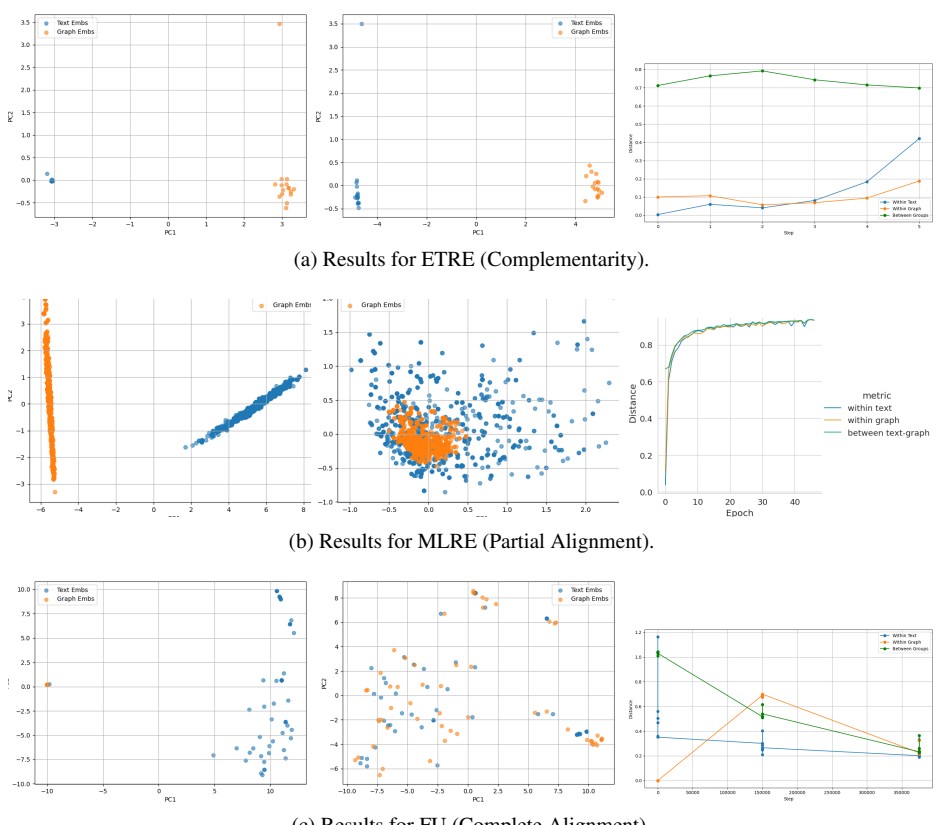

(a) Results for ETRE (Complementarity).

(b) Results for MLRE (Partial Alignment).

(c) Results for FU (Complete Alignment).

Figure 3: Representative examples of (1) PCA visualizations at initial and final epochs; (2) distances within text (blue), within graph (yellow), and between the two (green).

# 5 Conclusion

We analyze how text and graph representations complement each other under a unified framework unified contrastive co-distillation (CoD). Across five relational reasoning tasks we selected, we observe a spectrum from complementarity to alignment, shaped by factors such as whether the graph encodes the target directly, whether nodes map to text spans, and whether reasoning is local or global. These findings improve our understanding of text-graph representation relations and offer practical insights into applying CoD in structured NLP tasks.

# 6 Acknowledgements

This work is supported by NSF DRK12-2405615 and NSF ITEST-2241670.

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

| Task | Goal | Input | Output | Example | K-Type |
|------|------|-------|--------|---------|--------|
| ETRE | Predict temporal relation between two events | Text passage + Syntactic graph and Time-aware graph | Relation label (e.g., BE-FORE/AFTER) | In: Atlanta nineteen ninety-six. A bomb <E1> blast </E1> shocks the Olympic games. One person is killed. January nineteen ninety-seven. Atlanta again. This time a bomb at an abortion clinic. More people are <E2> hurt </E2>. Out: Event E1 took place BEFORE Event E2. | Episodic |
| MLRE | Predict semantic relation between entities | Text passage + Dependency graph | Relation label (e.g. sibling) | In: The <E1> wood </E1> is used as fuel and to make posts for <E2> fences </E2>. Out: The relation between E1 and E2: material used | Episodic |
| FU | Predict token relationships in scanned forms | OCR tokens with layout info | Label over token pairs | We present an example in Figure 5 | Episodic |
| RPP | Predict reasoning path over the KG for a question. | Question + KG subgraph | Reasoning Path | In: Question: What was Elie Wiesel's father's name? KG: Elie Wiesel <E1> \| <E1> book.author.book_editions_published <E2> \| <E3> people.person.gender <E4> ... | Static |
| KBQA entity-ranking | Extract answers from a KG for a question | | Ranked list of candidate entities | Out: Reasoning Pattern Type: T2 — The answer is located a single-hop away from the two constraints. Entities ranked: <E6>, <E4>, ... | |

Table 2: For each task, we state the goal, the input/output format, an illustrative example, and the graph construction method. We also distinguish between tasks grounded in *episodic* knowledge (context-dependent and document-specific), and those involving *static* knowledge (holds independently of context) in the Knowledge(K)-Type column.

# A    Task suite details

## A.1    Task illustrations

See Table 2.

## A.2    Data processing for reasoning pattern prediction and KBQA entity-ranking

| RP | Illustration | Definition | Example Question | S-expression |
|----|--------------|------------|------------------|--------------|
| T-0 | | A single-hop path from the con-straint to the answer. | What is the name of money in Brazil? | (JOIN (R location.country.currency_used) m.015fr) |
| T-1 | | A two-hop path from the constraint to the answer. | Where does the Queen of Denmark live? | (JOIN (R people.place_lived.location) (JOIN (R people.person.places_lived) m.0g2kv)) |
| T-2 | | Two single-hop paths arising from two different constraints and con-verging to the same answer. | What was Elie Wiesel's father's name? | (AND (JOIN people.person.gender m.05zppz) (JOIN (R people.person.parents) m.02vsp)) |
| T-3 | | Two paths (one single-hop and an-other two-hop) arising from two dif-ferent constraints and converging to the same answer. | Where did Joe Namath at-tend college? | (AND (JOIN common.topic.notable_types m.01y2hnl) (JOIN (R educa-tion.education.institution) (JOIN (R people.person.education) m.01p_3k))) |
| T-4 | | Two two-hop paths arising from two different constraints and converging to an intermediate common node be-fore reaching the answer. | Who does Zach Galifi-anakis play in The Hang-over? | (JOIN (R film.performance.character) (AND (JOIN film.performance.film m.0n3xxpd) (JOIN (R film.actor.film) m.02_0d2))) |

Table 3: Reasoning patterns with their corresponding definitions, example questions, and S-expressions.

We use the WebQSP dataset [Yih et al., 2016] for our two KBQA related experiments, i.e. reasoning pattern prediction and entity-ranking. An exploratory analysis of WebQSP highlighted a significant overlap of relations and classes across the train and test splits. Subsequently, we employed the approach of Jiang and Usbeck [2022] to obtain development and test splits that characterize different generalization levels in equal proportion. The three generalization levels for KBQA tasks include i.i.d, compositional, and zero-shot.

The i.i.d. case implies that the questions observed during inference follow similar logical templates to those during training; for example the questions "Who was the author of Oliver Twist?" and "Who wrote Pride and Prejudice?" follow similar logical templates. We contrast this with the compositional case, where questions in the test split operate over the same set of relations that were present in the

| RP | Illustration | i.i.d. | Comp | Z.S. | Total |
|---|---|---|---|---|---|
| T-0 | | 50.3 | 0.0 | 49.7 | 54.5 |
| T-1 | | 37.3 | 44.3 | 18.4 | 23.5 |
| T-2 | | 17.1 | 47.1 | 35.7 | 5.2 |
| T-3 | | 83.3 | 6.7 | 10.0 | 2.2 |
| T-4 | | 12.8 | 81.5 | 5.6 | 14.5 |
| ALL | | 40.8 | 24.9 | 34.3 | 100.0 |

Table 4: Distribution of reasoning patterns over the generalization splits (i.i.d., compositional (Comp), zero-shot (Z.S.)) of our modified WebQSP dataset.

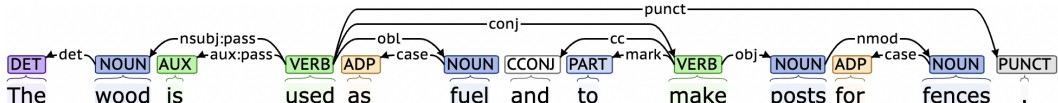

Figure 4: Example depicting the supplemental information provided by the *dependency tree*. The entities of interest are wood and fences, having the relationship material_used. The path *wood ← used → make → posts → fences* elicits this relationship.

training set (such as the "written-by" relation), but different logical templates. For example, the questions "Who wrote Pride and Prejudice?" and "Who wrote both The Talisman and It?" require reasoning over the same relation "written-by" but follows different reasoning paths, since the former involves only one constraint or entity, whereas the latter involves two. Finally, questions in the zero-shot split operate over new or unseen relations that were not present in the training dataset. For example, the questions "Who wrote Pride and Prejudice?" and "Who directed Pride and Prejudice in 2005?" involves different relations, i.e. "written-by" and "directed-by" respectively. We defer the readers to past work [Gu et al., 2021, Jiang and Usbeck, 2022, Dutt et al., 2023] for a more thorough description of the different generalization splits.

We characterize the complexity of the reasoning pattern to answer a given KBQA question based on Dutt et al. [2023]. Given the modified version of WebQSP dataset, we identify the following five reasoning patterns that accounted for $\geq 97\%$ of the dataset across all splits. We describe the different reasoning patterns in Table 3 and outline their distribution in the our modified WebQSP dataset in Table 4.

To accommodate the input length constraints of models like T5, we simplify the representation of knowledge base entities in the linearized graph input. Instead of using full entity identifiers (e.g., m.02896), we assign short, unique placeholder tokens (e.g., <E1>, <E2>) to each entity as a part of the tokenizer vocabulary. This helps reduce the input sequence length and avoids unwanted subword tokenization. In addition, we ensure that these placeholder tokens are assigned consistently across modalities: the same entity is represented as node $v_i$ in the graph and as token <Ei> in the linearized text.

## A.3 MLRE dependency parsing illustration

See Figure 4.

## A.4 FU example

We adapt an example to showcase the FU task from Nourbakhsh et al. [2024] in Figure 5.

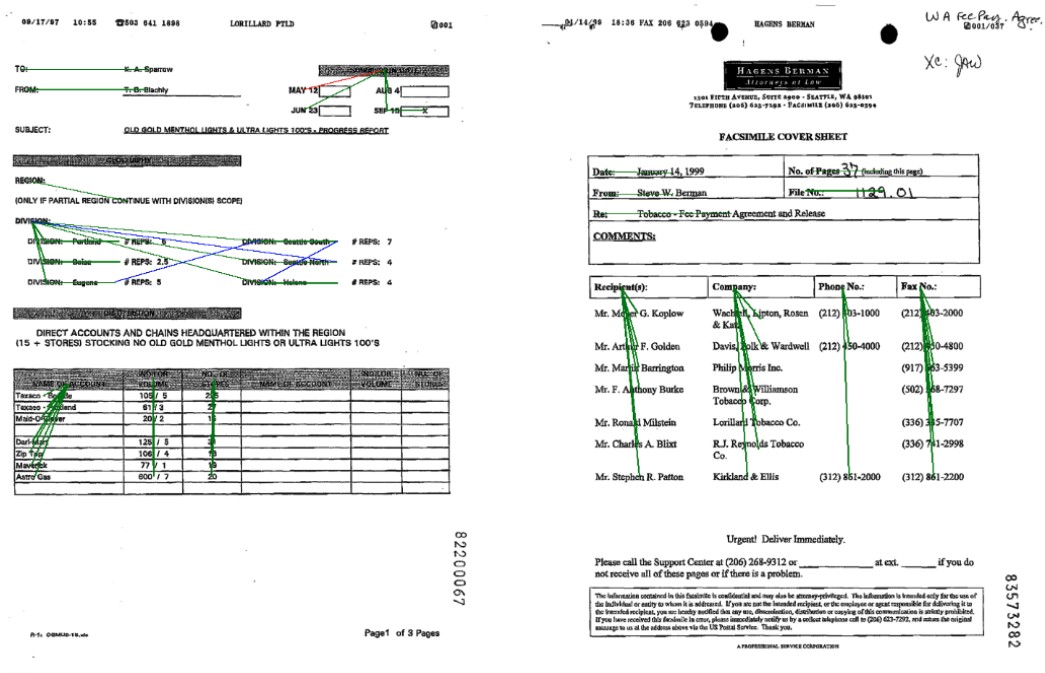

Figure 5: An example of FU task from the FUNSD dataset, adapted from Nourbakhsh et al. [2024]. Green links show correct predictions. Red links show false negatives. Blue links show false positives.

| Task | Text model | Graph model | Loss function | Metric |
|---|---|---|---|---|
| ETRE | RoBERTa[1] | {1,2,3}-layer RGAT[4] | cross-entropy (CE) | weighted F1 |
| Form understanding | RoBERTa[1] | 2-layer RGAT[4] | binary CE | F1 |
| MLRE | mBERT-base[2] | 2-layer RGCN[5] | CE | macro F1 |
| Reasoning pattern prediction | T5-base[3] | 2-layer RGCN[5] | CE | macro F1 |
| KBQA entity-ranking | T5-base[3] | 2-layer RGCN[5] | binary CE | Hits@$K$[6] |

Table 5: Model configurations, training objectives, and evaluation metrics for each task. The text and graph model backbones listed in this table are used for the primary results in Table 1.

# B Task experiments details

We present the experimental details for different tasks. In Table 5, we outline the loss function that we are optimizing, the corresponding evaluation metric, and the backbone architectures used for the primary results reported in Table 1: the transformer model that encodes the textual information, and the specific GNN architecture that encodes the graph information. In Table 6, we provide hyperparameters values for our experiments. We also present statistics on the task suite datasets and training times in Table 7. All datasets we used are publicly available, and we follow the licensing terms and intended use of each.

| Task | LR | Batch size | Drop out | Temp. | Max input len | GNN layers | GNN hidden dim |
|---|---|---|---|---|---|---|---|
| ETRE (TDDMan) | 1e-5 | 16 | 0.1 | 0.1 | – | 2 | 256 |
| ETRE (TDDAuto) | 1e-5 | 32 | 0.1 | 0.04 | – | 3 | 256 |
| ETRE (TB-Dense) | 1e-5 | 32 | 0.1 | 0.9 | – | 1 | 256 |
| MLRE | 1e-5 | 16 | 0.2 | 0.1 | 512 | 2 | 768 |
| Reasoning pattern prediction | 5e-5 | 6 | 0.2 | 0.1 | 512 | 2 | 768 |
| KBQA entity-ranking | 5e-5 | 4 | 0.2 | 0.1 | 1024 | 2 | 768 |
| Form understanding | *Same settings as in* Nourbakhsh et al. [2024] | | | | | | |

Table 6: Hyperparameters used across tasks. Temperature refers to $\tau$ in CoD. All experiments use a shared space dimension of 2048.

| Task | Dataset | Train | Test | Number of labels | Training time |
|---|---|---|---|---|---|
| **ETRE** | TDDMan | 4,000 | 1,500 | 5 | 28 min |
| | TDDAuto | 32,609 | 4,258 | 5 | 3h 40min |
| | TB-Dense | 4,032 | 1,427 | 6 | 26 min |
| **MLRE** | REDFM (en) | 8,504 | 1,235 | 32 | 6h 7min |
| | REDFM (es) | 5,194 | 733 | 32 | 2h 30min |
| | REDFM (fr) | 5,452 | 975 | 32 | 3h 14min |
| | REDFM (de) | 5,909 | 811 | 32 | 2h 46min |
| | REDFM (it) | 4,597 | 1,086 | 32 | 2h 38min |
| **Reasoning pattern prediction** | WebQSP | 3,014 | 1,343 | 5 | 1h |
| **KBQA answer-ranking** | WebQSP | 3,014 | 1,343 | Number of gold answers | 3h |
| **Form understanding** | SROIE | 626 | 347 | 4 | 10h |
| | FUNSD | 149 | 50 | 4 | 4h 36min |
| | CORD | 800 | 100 | 30 | 17h 47min |

Table 7: Task suite statistics and training times. We train for 1000 epochs for form understanding.

# C   Extended CoD results

To further demonstrate the robustness and generality of CoD, we apply it to new model combinations on two representative tasks: reasoning pattern prediction and ETRE (Table 8). We also demonstrate additional CoD performance across each language data for MLRE in Table 9.

# D   Full visualization results across tasks

## D.1   ETRE results

See Figure 8, Figure 6 and Figure 7 for results on TDDMan, TimeBank-Dense and TDDAuto datasets, respectively. See Figure 9 for results on TDDMan dataset when no CoD is applied.

## D.2   MLRE results

See Figure 10 for PCA plots, and Figure 11 for cosine similarity and distance metrics results.

---

[1]Liu et al. [2019]

[2]Devlin et al. [2019]

[3]Raffel et al. [2023]

[4]Busbridge et al. [2019]

[5]Schlichtkrull et al. [2017]

[6]$K$ indicates the number of correct answers for an instance.

| (a) Reasoning pattern prediction | | | | |
|---|---|---|---|---|
| Text encoder | Graph encoder | Hybrid (CoD) | Text only | Graph only |
| T5 | RGCN | **0.6190** | 0.5700 | 0.5840 |
| T5 | RGAT | **0.6120** | 0.5700 | 0.4966 |
| BERT | RGCN | **0.5999** | 0.5835 | 0.5840 |
| BERT | RGAT | **0.5956** | 0.5835 | 0.4966 |
| GPT-2 | RGCN | **0.6022** | 0.5614 | 0.5840 |
| GPT-2 | RGAT | **0.6049** | 0.5614 | 0.4966 |

| (b) Event temporal relation extraction (ETRE) | | | |
|---|---|---|---|
| Text encoder | Graph encoder | Hybrid (CoD) | Text only |
| *TDDMan* | | | |
| BERT | GCN | 0.411 | **0.447** |
| BERT | RGCN | 0.384 | **0.447** |
| BERT | RGAT | **0.481** | 0.447 |
| RoBERTa | GCN | 0.435 | **0.445** |
| RoBERTa | RGCN | **0.452** | 0.445 |
| RoBERTa | RGAT | **0.551** | 0.445 |
| *TDDAuto* | | | |
| BERT | GCN | **0.631** | 0.624 |
| BERT | RGCN | **0.647** | 0.624 |
| BERT | RGAT | **0.683** | 0.624 |
| RoBERTa | GCN | **0.748** | 0.689 |
| RoBERTa | RGCN | 0.665 | **0.689** |
| RoBERTa | RGAT | **0.771** | 0.689 |
| *TB-Dense* | | | |
| BERT | GCN | **0.790** | 0.775 |
| BERT | RGCN | **0.782** | 0.775 |
| BERT | RGAT | **0.810** | 0.775 |
| RoBERTa | GCN | **0.805** | 0.767 |
| RoBERTa | RGCN | **0.847** | 0.767 |
| RoBERTa | RGAT | **0.856** | 0.767 |

Note that we did not record numbers for the graph-only approach because the graph approach for this task yields incredibly poor results without the incorporation of linear transformers [Yao et al., 2024].

Table 8: Additional results for (a) Reasoning pattern prediction and (b) ETRE using different text and graph encoder backbones. CoD consistently improves over baselines across all combinations in Reasoning pattern prediction, and improves 78% of the times across all 18 cases for ETRE. These results demonstrate CoD's generality across diverse model architecture combinations.

| Language | Text only | Graph only | Hybrid + CoD | Hybrid + no-CoD |
|---|---|---|---|---|
| de | **80.41** $\pm$ 0.61 | 47.13 $\pm$ 2.76 | 80.35 $\pm$ 0.71 | 79.55 $\pm$ 0.40 |
| en | **85.94** $\pm$ 1.41 | 52.21 $\pm$ 0.56 | 84.57 $\pm$ 2.25 | 84.74 $\pm$ 1.07 |
| es | **80.49** $\pm$ 0.61 | 51.21 $\pm$ 1.47 | 76.64 $\pm$ 1.09 | 80.26 $\pm$ 0.44 |
| fr | 77.47 $\pm$ 0.73 | 45.62 $\pm$ 1.60 | **78.80** $\pm$ 0.58 | 78.31 $\pm$ 0.78 |
| it | 74.25 $\pm$ 0.36 | 46.61 $\pm$ 1.98 | 72.67 $\pm$ 1.40 | **74.76** $\pm$ 1.02 |
| Avg | **79.71** $\pm$ 3.95 | 48.55 $\pm$ 3.21 | 78.61 $\pm$ 4.17 | 79.53 $\pm$ 3.32 |

Table 9: F1 score results on MLRE task for the RED[fm] dataset.

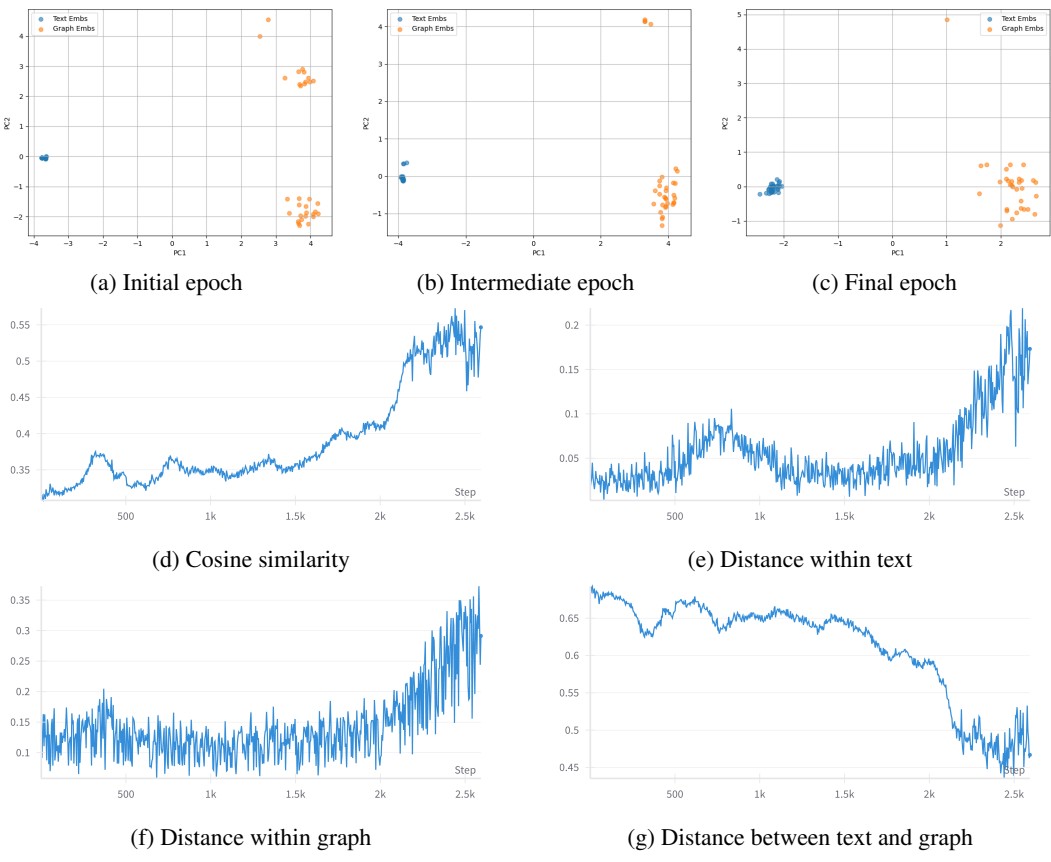

(a) Initial epoch
(b) Intermediate epoch
(c) Final epoch

(d) Cosine similarity
(e) Distance within text

(f) Distance within graph
(g) Distance between text and graph

Figure 6: Results for ETRE on the TimeBank-Dense dataset.

## D.3 RPP results

See Figure 13 and Figure 12 for Reasoning Pattern Prediction task with and without CoD applied, respectively.

## D.4 FU results

See Figure 16, Figure 14, and Figure 15 for results on CORD, SROIE, and FUNSD datasets, respectively.

## D.5 KBQA entity-ranking results

See Figure 17 and Figure 18 for results for KBQA entity-ranking with and without CoD applied, respectively.

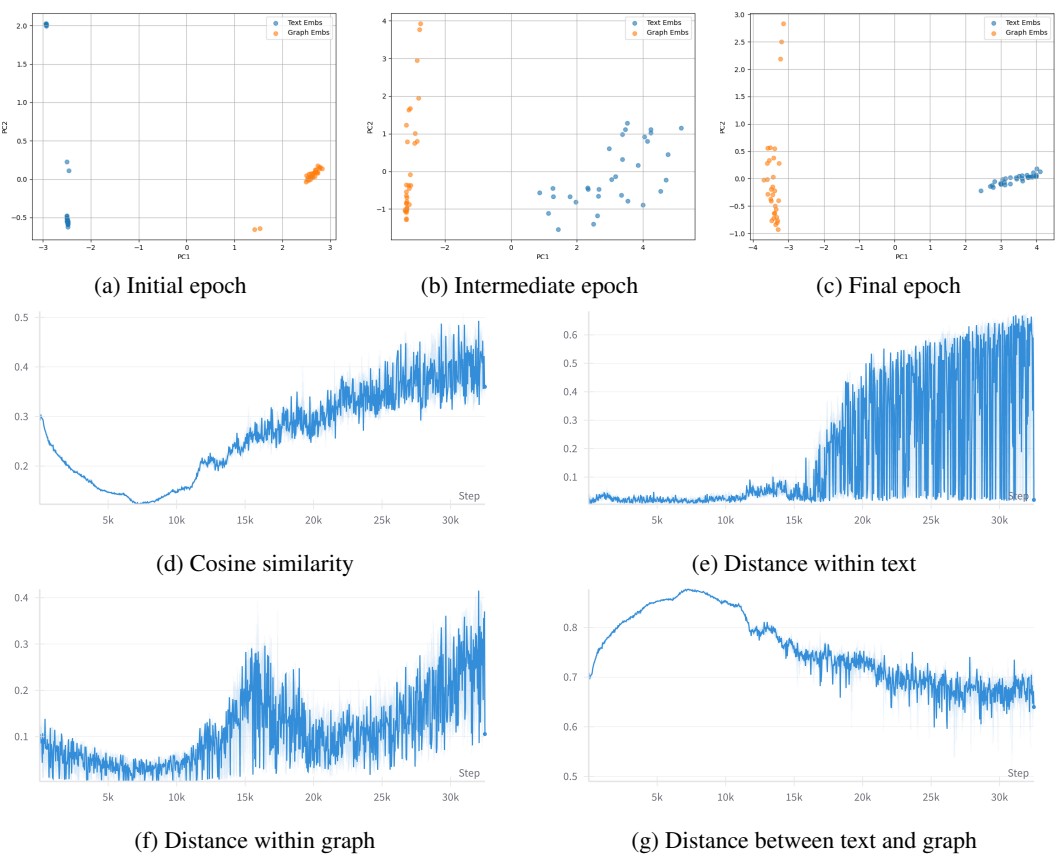

(a) Initial epoch      (b) Intermediate epoch      (c) Final epoch

(d) Cosine similarity      (e) Distance within text

(f) Distance within graph      (g) Distance between text and graph

Figure 7: Results for ETRE on the TDDAuto dataset.

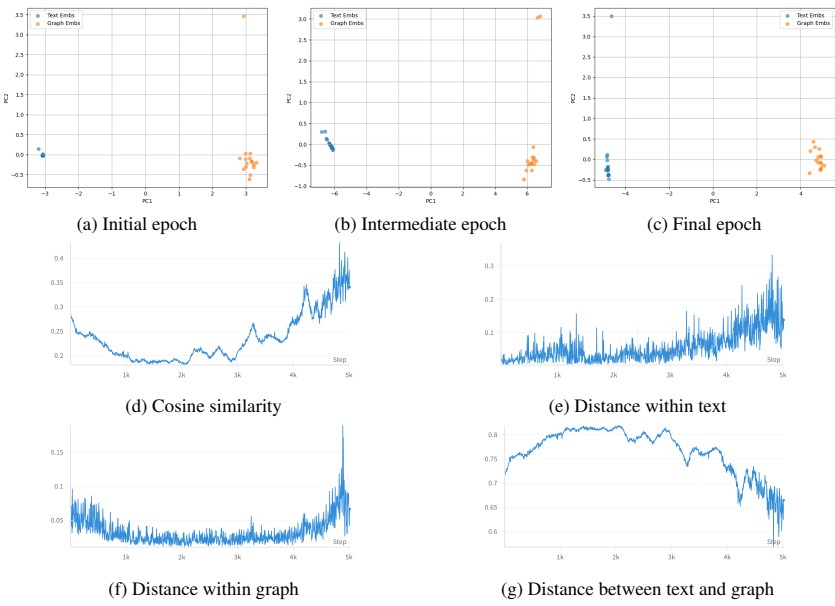

(a) Initial epoch      (b) Intermediate epoch      (c) Final epoch

(d) Cosine similarity      (e) Distance within text

(f) Distance within graph      (g) Distance between text and graph

Figure 8: Results for ETRE on the TDDMan dataset. PCA visualizations (top) at initial, intermediate, and final training stages, and corresponding distance-based metrics (bottom).

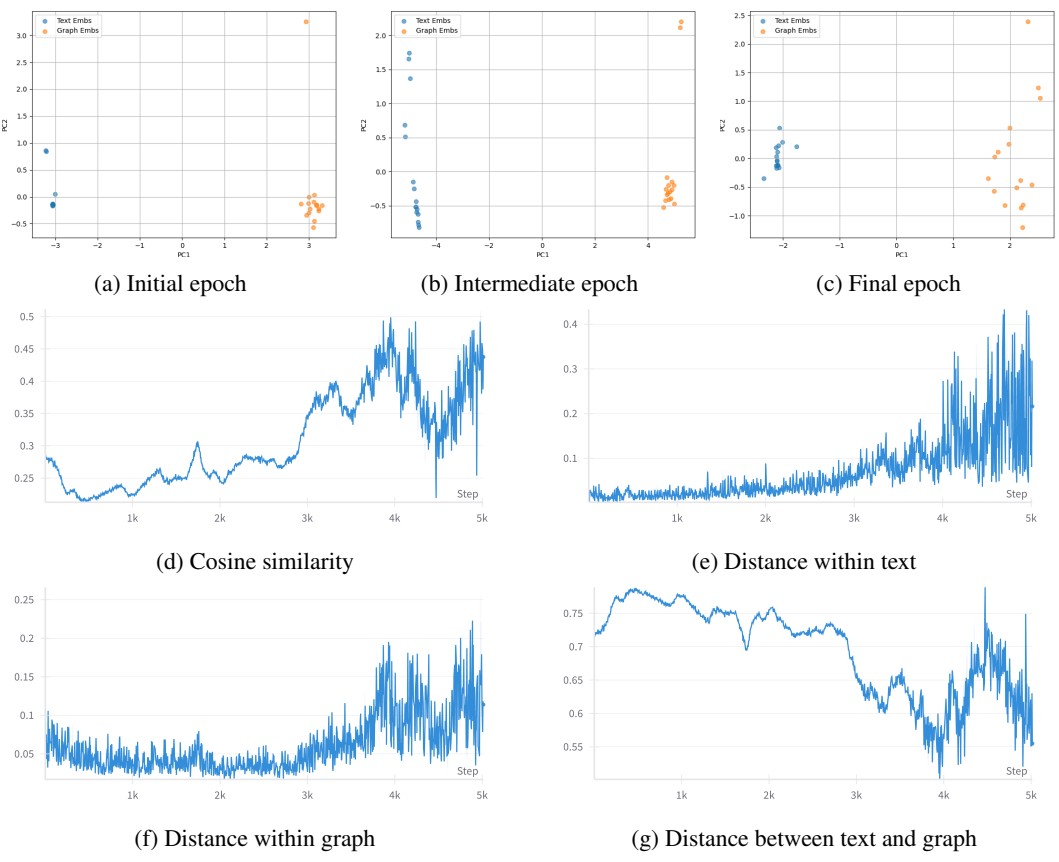

(a) Initial epoch      (b) Intermediate epoch      (c) Final epoch

(d) Cosine similarity      (e) Distance within text

(f) Distance within graph      (g) Distance between text and graph

Figure 9: Results for ETRE on the TDDMan dataset when no CoD is applied.

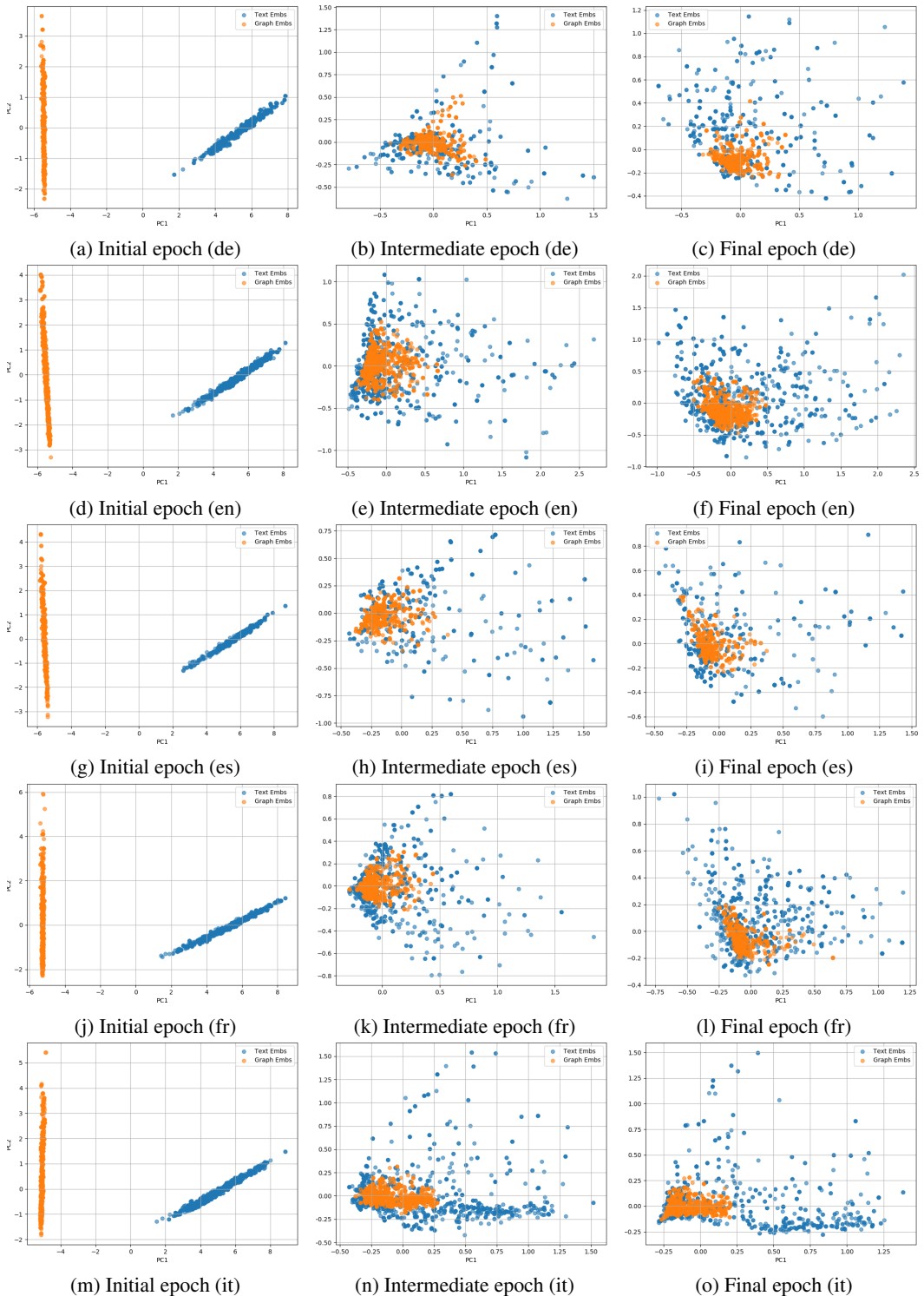

Figure 10: PCA plots for MLRE across the different languages in the RED$^{fm}$ dataset.

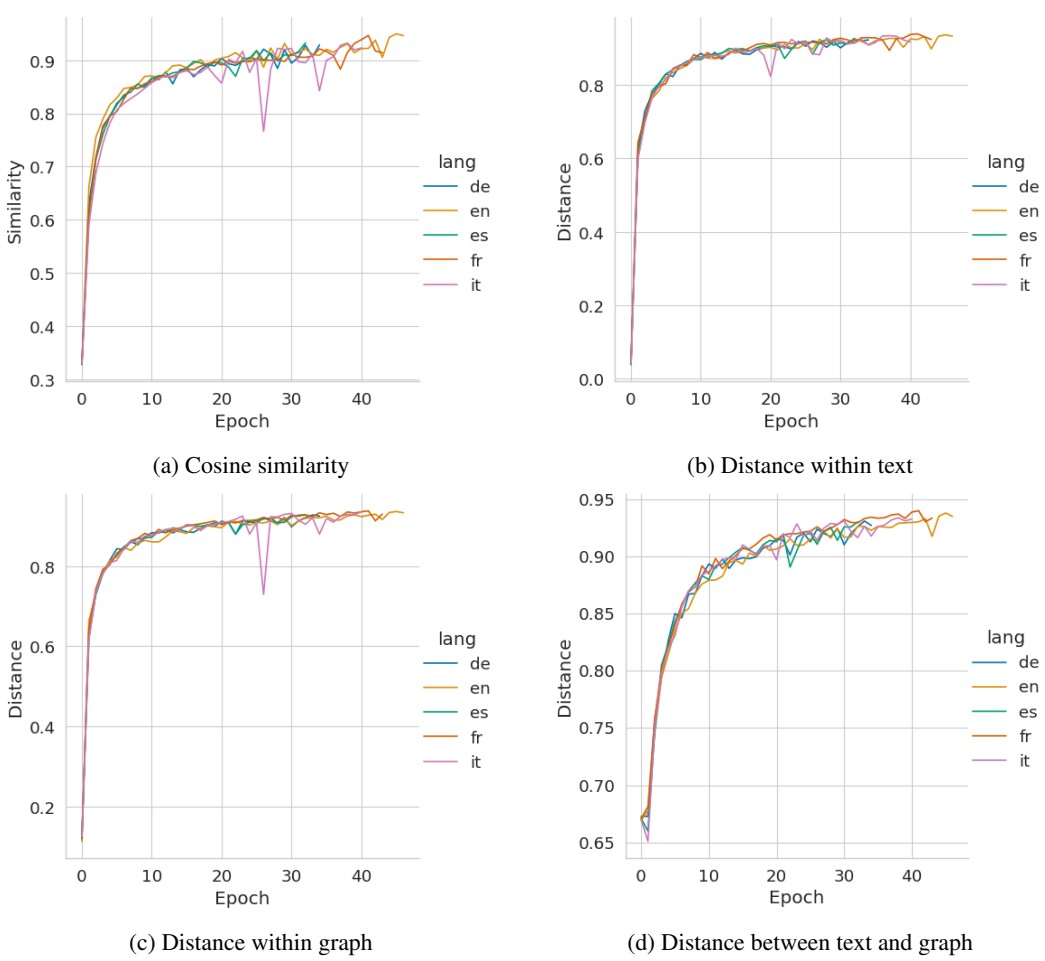

(a) Cosine similarity

(b) Distance within text

(c) Distance within graph

(d) Distance between text and graph

Figure 11: Cosine similarity and distance results for MLRE on the RED$^{\text{fm}}$ dataset.

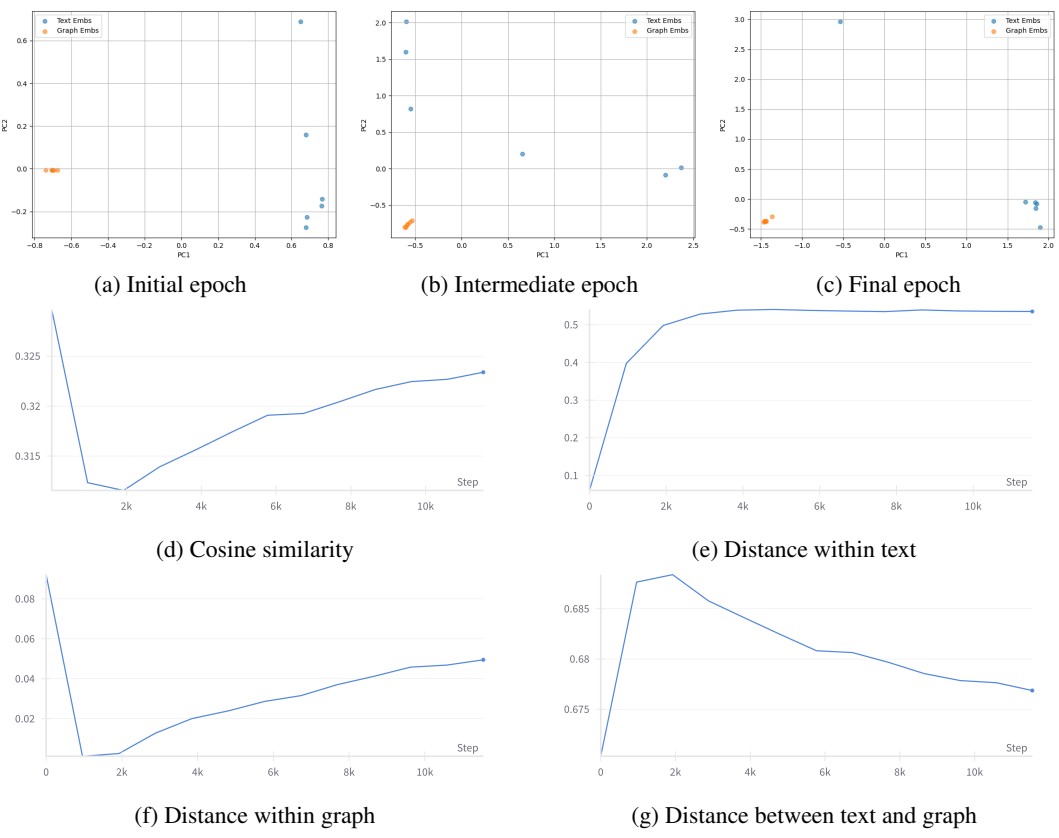

(a) Initial epoch      (b) Intermediate epoch      (c) Final epoch

(d) Cosine similarity      (e) Distance within text

(f) Distance within graph      (g) Distance between text and graph

Figure 12: Results for reasoning pattern prediction on the WebQSP dataset when no CoD is applied.

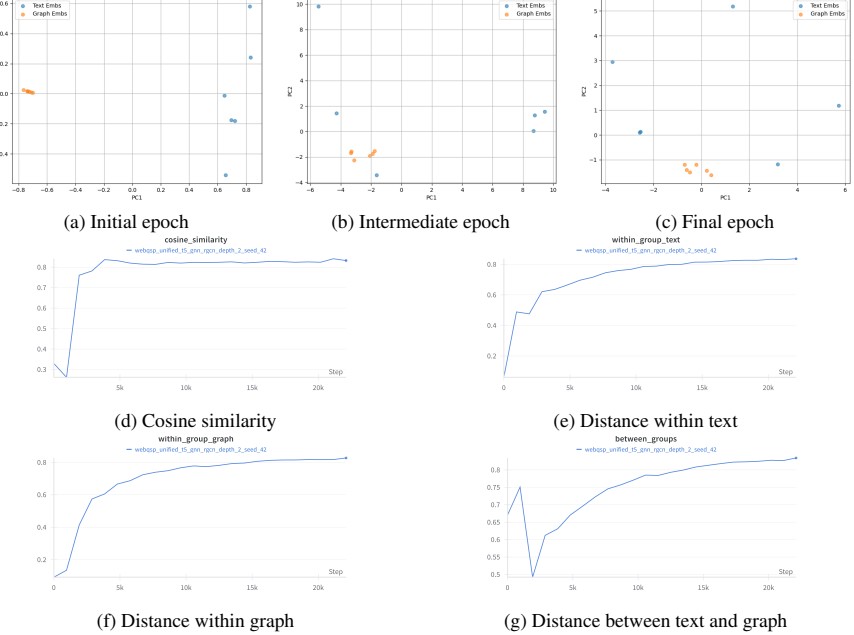

(a) Initial epoch      (b) Intermediate epoch      (c) Final epoch

(d) Cosine similarity      (e) Distance within text

(f) Distance within graph      (g) Distance between text and graph

Figure 13: Results for reasoning pattern prediction on the WebQSP dataset.

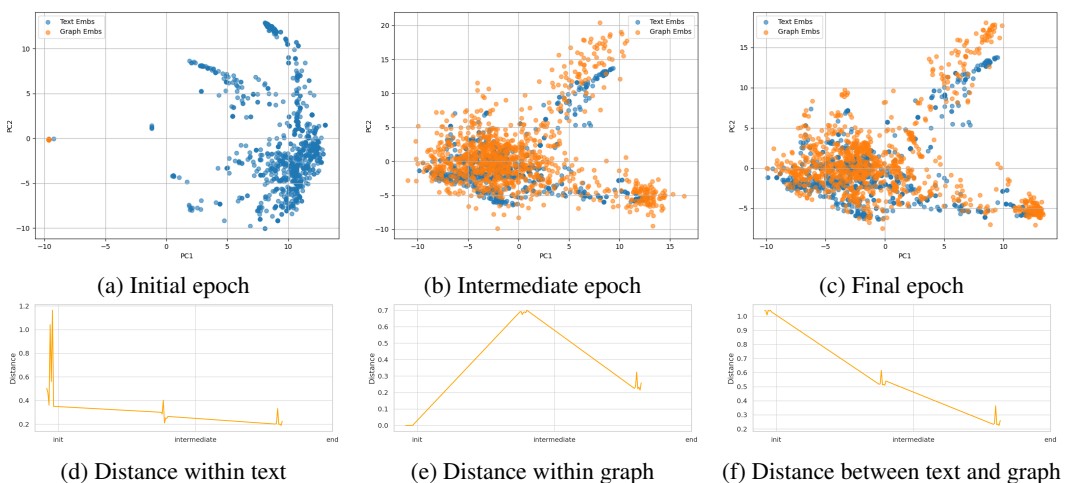

(a) Initial epoch      (b) Intermediate epoch      (c) Final epoch

(d) Distance within text      (e) Distance within graph      (f) Distance between text and graph

Figure 14: Results for form understanding on the SROIE dataset.

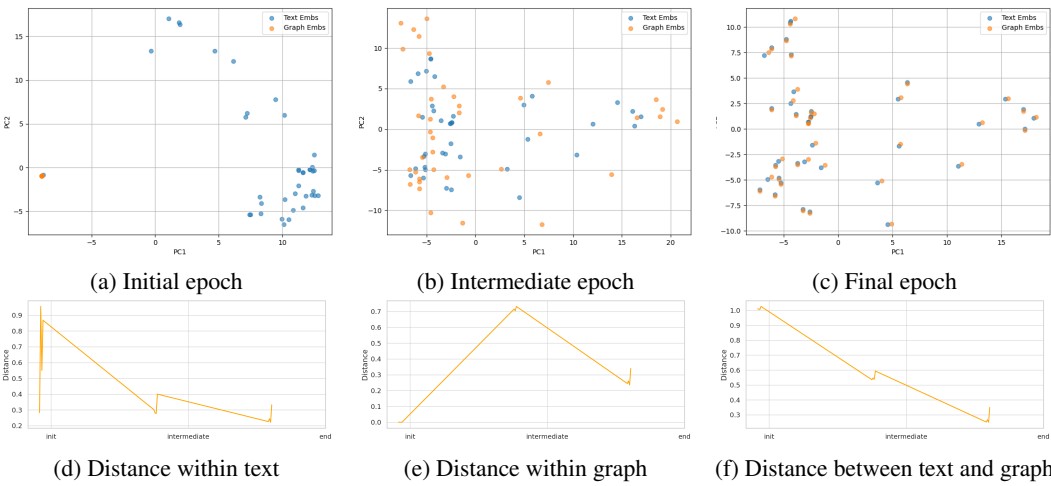

(a) Initial epoch      (b) Intermediate epoch      (c) Final epoch

(d) Distance within text      (e) Distance within graph      (f) Distance between text and graph

Figure 15: Results for form understanding on the FUNSD dataset.

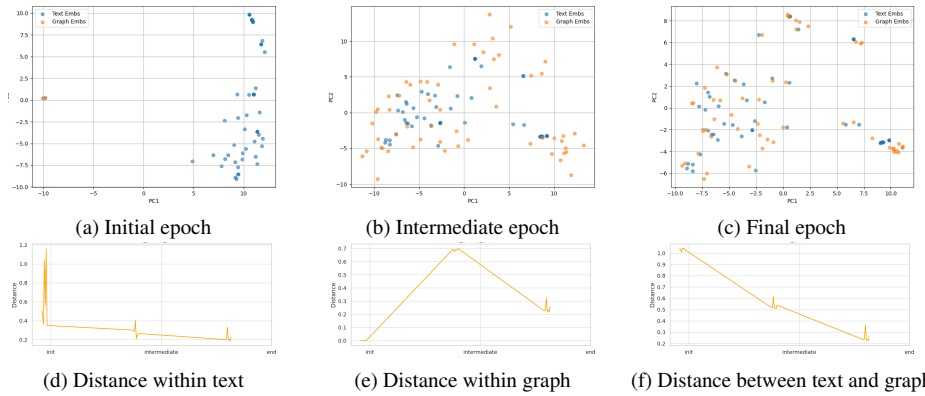

(a) Initial epoch      (b) Intermediate epoch      (c) Final epoch

(d) Distance within text      (e) Distance within graph      (f) Distance between text and graph

Figure 16: Results for form understanding on the CORD dataset.[3]

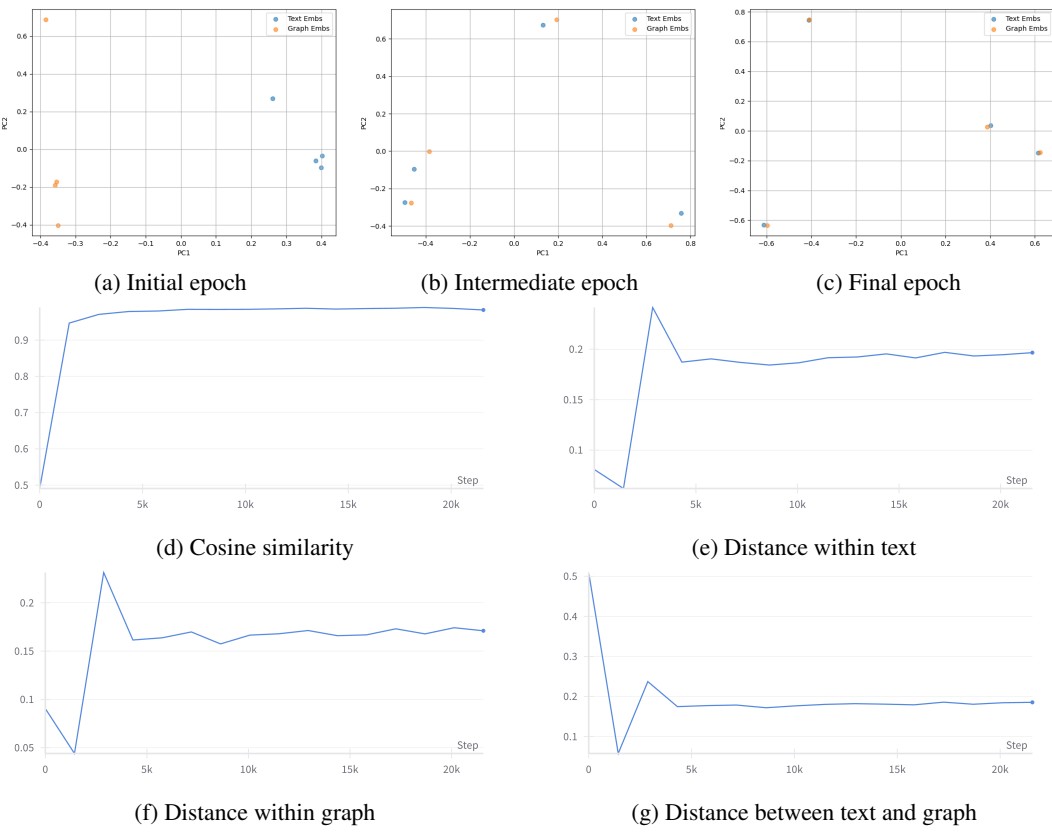

(a) Initial epoch      (b) Intermediate epoch      (c) Final epoch

(d) Cosine similarity      (e) Distance within text

(f) Distance within graph      (g) Distance between text and graph

Figure 17: Results for KBQA entity-ranking on the WebQSP dataset.

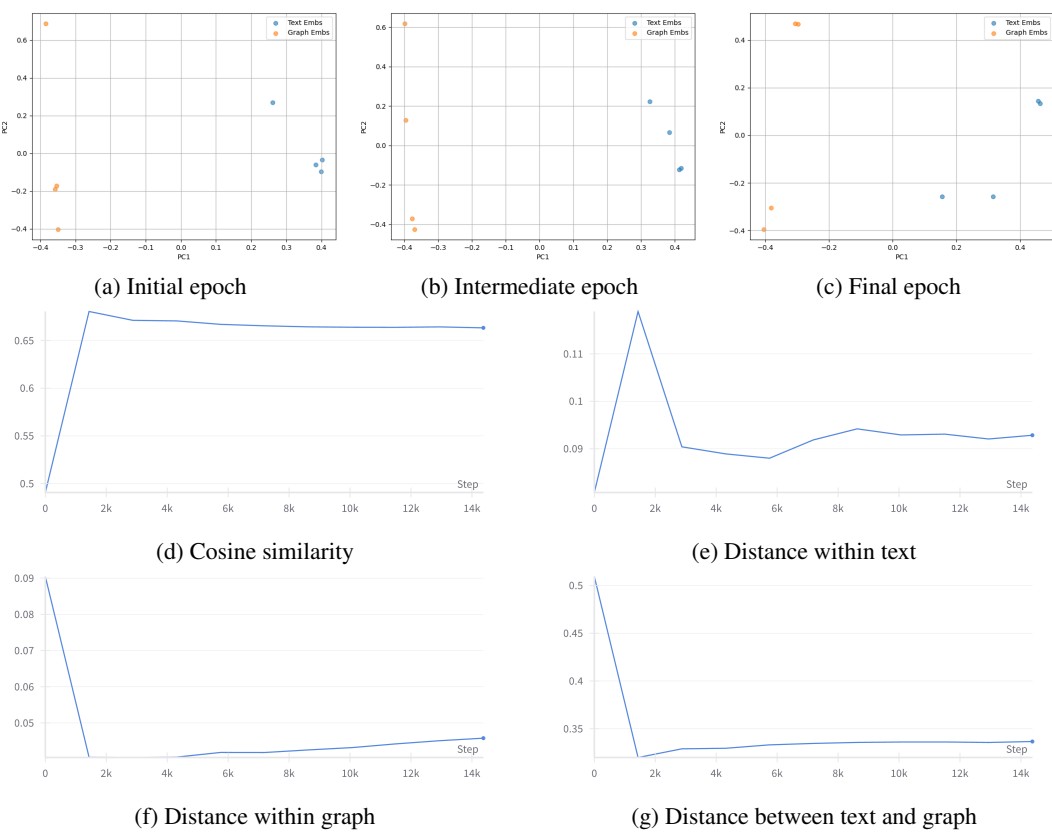

(a) Initial epoch      (b) Intermediate epoch      (c) Final epoch

(d) Cosine similarity      (e) Distance within text

(f) Distance within graph      (g) Distance between text and graph

Figure 18: Results for KBQA entity-ranking on the WebQSP dataset when no CoD is applied.

