# OpenReview forum: "R²-CoD: Understanding Text-Graph Complementarity in Relational Reasoning via Knowledge Co-Distillation"
_NeurIPS.cc/2025/Workshop/UniReps — UniReps2025_

### Official Review · Reviewer_rsAq · 2025-09-13
**Good study on text-graph complementarity in relational reasoning**

**Confidence:** 3

**Review:**

The paper studies how text and graph representations complement or align in relational reasoning tasks. The authors propose a unified framework (R2-CoD) that applies contrastive knowledge co-distillation (CoD) to enable mutual teaching between text and graph encoders.
They evaluate across five tasks (ETRE, MLRE, RPP, FU, KBQA), covering a spectrum from complementarity (ETRE) to near-complete alignment (FU, KBQA) of information.


The work provides an analysis of representational dynamics, analysing when and why text–graph integration is beneficial and providing an explanation as to why some tasks have greater improvements than others when text-graph with CoD is used.


**Quality**: the paper is of good quality. The analysis are complete and cover a diverse set of relational reasoning tasks and datasets. The visualizations with PCA and similarity metrics at different epochs add interpretability to the results, which I really liked.


**Clarity**: the clarity is good, the paper is well written and very easy to follow. The figures are well done and help in understanding the text.


**Originality**: the main originality is the systematic study of when and why graphs can help beyond text in NLP relational reasoning. To the best of my knowledge, this is novel work.

**Significance**: the results have practical significance because they show that task characteristics determine whether text–graph integration is truly beneficial. I believe this work can serve as a practical guideline for other researchers interested in relational reasoning to better shape their research. I think the significance is solid for the community.


Overall, this work offers a useful and well done analysis of text–graph complementarity in relational reasoning with a unified CoD approach. Its strength lies in the interpretability and systematic evaluation.

**Score:**

4

**Topic Fit:**

3

---

### Official Review · Reviewer_BpFz · 2025-09-15

**Confidence:** 4

**Review:**

# Summary

This paper introduces R^2-CoD, a framework for analyzing the alignment between text and graph representations in relational reasoning tasks. The key idea is to train text and graph encoders jointly with both a task loss and a contrastive co-distillation (CoD) loss, while projecting each modality into a shared space for comparison. The authors apply the approach to five tasks (ETRE, MLRE, FU, RPP, KBQA), demonstrating a spectrum of outcomes when it comes to alignment. They report consistent performance improvements and provide visualization-based insights into representation dynamics.

# Strengths

* Clarity: The paper is clearly written and well-structured.
* Conceptual framing: R^2-CoD is an interesting lens for thinking about multimodal reasoning and representation alignment.

# Weaknesses / Limitations

* Possible confounding effect of CoD: It is unclear how much of the observed effects reflects inherent task properties vs. CoD as a regularizer.
* Somewhat limited coverage: Additional tasks/datasets would strengthen the paper's conclusion.

# Questions/Suggestions

* Provide, or emphasize, quantitative criteria for classifying complementarity vs. alignment (e.g., thresholds on similarity).
* Clarify, or emphasize, why CoD sometimes enforces strong alignment and sometimes does not. Is this task-specific or an artifact of the CoD?
* Add an ablation to probe the confounding effect of CoD (e.g., frozen encoders, varying the weight of the CoD term).
* Run the analysis on more tasks.

**Score:**

3

**Topic Fit:**

3

---

### Official Review · Reviewer_7XxF · 2025-09-16
**R2-CoD - understanding text graph complementarity via knowledge co-distillation**

**Confidence:** 1

**Review:**

The paper addresses the question of how text and graph modalities complement or align in relational reasoning tasks. Using a unified architecture with contrastive knowledge co-distillation (CoD), it evaluates five representative tasks:
- event temporal relation extraction
- multilingual relation extraction
- form understanding
- KBQA
- relation path prediction

Results show consistent improvements from hybrid text+graph models. Paper provides representation level analyses using cosine similarity, PCA, distance metrics.

Strengths:
- R2-COD setup is clean, modular, and task agnostic. It can generalize across tasks.
- Goof empirical study: Includes PCA visualizations, cosine similarity trends, and comparisons across text-only, graph-only, hybrid, and hybrid+CoD baselines.

Weakness:
- Paper can theoretically further detail why complementarity/alignment arises under CoD.
- Experiments focus on a small set of datasets per task. Broader validation can further strengthen the paper.

**Score:**

3

**Topic Fit:**

2